


# Reconstruction and analysis of erythemal UV radiation time series from Hradec Králové (Czech Republic) over the past 50 years

Klára Čížková[1,2], Kamil Láska[1], Ladislav Metelka[2], Martin Staněk[2]

[1] Department of Geography, Faculty of Science, Masaryk University, Brno, 611 37, Czech Republic
[2] Solar and Ozone Observatory, Czech Hydrometeorological Institute, Hradec Králové, 500 08, Czech Republic

*Correspondence to*: Klára Čížková (cizkova.klara@hotmail.com)

**Abstract.** This paper evaluates the variability of erythemal ultraviolet (EUV) radiation from Hradec Králové (Czech Republic) in the period 1964–2013. The EUV radiation time series was reconstructed using a radiative transfer model and additional empirical relationships with the root mean square error of 9.9 %. The reconstructed time series documented the increase in EUV radiation doses in the 1980s and the 1990s (up to 15 % per decade), which is linked to the steep decline in total ozone (10 % per decade). The changes of cloud cover were the major factor affecting the EUV radiation doses especially in the 1960s, 1970s, and at the beginning of the new millennium. The mean annual EUV radiation doses in the decade 2004–2013 declined by 5 %. The factors affecting the EUV radiation doses differed also according to the chosen integration period (daily, monthly, and annually): solar zenith angle was the most important for daily doses, cloud cover for their monthly means, and the annual means of EUV radiation doses were most influenced by total ozone column. The number of days with very high EUV radiation doses increased by 22 % per decade, the increase was statistically significant in all seasons except autumn. The occurrence of the days with very high EUV doses was influenced mostly by low total ozone column (82 % of days), clear-sky or partly cloudy conditions (74 % of days) and by increased surface albedo (19 % of days). The principal component analysis documented that the occurrence of days with very high EUV radiation doses was much affected by the positive phase of North Atlantic Oscillation with an Azores High promontory reaching over central Europe. In the stratosphere, a strong Arctic circumpolar vortex and also the meridional inflow of ozone-poor air from the south-west were favourable for the occurrence of days with very high EUV radiation doses. This is the first analysis of the relationship between the high EUV radiation doses and macro-scale circulation patterns, and therefore more attention should be given also to other dynamical variables that may affect the solar UV radiation on the Earth surface.

## 1 Introduction

Solar ultraviolet (UV) radiation causes a wide variety of environmental and health effects, including the deceleration of the rate of photosynthesis, damage of DNA structures, and the increased risk of skin cancer in humans (e.g. Diffey, 1991; Caldwell, 2007). The scientific interest in the solar UV radiation has significantly increased since the 1980s, when it was discovered that the many adverse environmental and health effects of the solar ultraviolet (UV) radiation have been



reinforced by the thinning of the ozone layer (e.g. Farman et al., 1985; Krzyścin and Borkowski, 2008). The effects of UV radiation on human skin differ according to the wavelength, so the erythema action spectrum (McKinlay and Diffey, 1987) was developed to model the susceptibility of human skin to sunburn. Therefore, it is important to assess the changes in the erythemal ultraviolet (EUV) radiation over a longer period of time, and relate them to processes and events that are involved

in the attenuation of solar radiation passing through the atmosphere (WHO, 2006).

Incident UV radiation intensity is affected by a broad range of atmospheric and environmental factors, of which the total ozone column (TOC) is arguably the most studied one. Ozone, together with molecular oxygen, absorbs all UV-C and a part of UV-B radiation, which are the most harmful to organisms (Bais, 2015). The changes in UV radiation are further modulated by cloudiness and cloud types, atmospheric aerosols, as well as solar zenith angle, and the altitude and ground

surface characteristics of the location (Kerr, 2005). According to numerous studies, atmospheric circulation and stratospheric temperature greatly affect the intensity of global ozone losses (e.g. Orsolini and Limpasuvan, 2001; Schnadt and Dameris, 2003; Hofmann et al., 2009). In the 1970s–1990s, TOC trends in central Europe declined by about 3–4 % per decade, especially in spring, while recovery of the ozone layer has been reported since the end of the 1990s (e.g. Krzyścin et al., 1998; Krzyścin and Borkowski, 2008; Vaníček et al., 2012; De Bock et al., 2014).

So far, the analyses of EUV radiation trends have been carried out in different locations across the Northern Hemisphere. A major increase of EUV radiation doses was observed in the 1980s and 1990s; and according to Ziemke et al. (2000), the trends ranged from 4.5 % per decade in the Mediterranean region to 8.6 % per decade in East Asia. The studies from central Europe reported an increase of EUV daily doses ranging from 2–17 % per decade (Rieder et al., 2008), but most commonly between 5–6 % per decade (e.g. Den Outer et al., 2010; Krzyścin et al., 2011). According to Rieder et al. (2008), two-thirds

of the ascertained trends were attributed to the changes in TOC, and the remaining one-third to the combined decrease of cloudiness and aerosol optical depth. In spite of the ozone layer recovery since the mid-1990s, the latter period does not reveal any statistically significant changes in EUV irradiation (Hadzimustafic et al., 2013). At the same time, an estimate of future trend in UV radiation remains unclear because it is often overlaid by the influence of cloudiness and aerosols, whose trends significantly vary at different European sites (Bais et al., 2015).

The solar UV radiation reaching the Earth's surface can be measured either by broadband or narrowband UV radiometers, or by spectrophotometers, like the Brewer spectrophotometer. Unlike the broadband radiometers, the Brewer spectrophotometer provides the most accurate UV radiation observations and is therefore considered a reference instrument (e.g. Gardiner et al., 1993; Anav et al., 1996; Hülsen et al., 2008). Nevertheless, the UV radiation measurements taken by the Brewer spectrophotometers are sparse and often cover only a short period of time, which complicates their incorporation into

the evaluation of long-term trends and effects on human health (Rieder et al., 2008). So as to assess the UV radiation variability in the past, time series reconstructions are essential. In central Europe, several UV radiation reconstructions have recently been performed (e.g. Krzyścin et al., 2004; Rieder et al., 2008; Den Outer et al., 2010); however, the reconstructed time series analyses do not extend to the last decade.



The EUV radiation time series presented in this study covers five decades (1964–2013) and uses radiative transfer modelling and the most precise EUV radiation measurements available at Hradec Králové (Czech Republic). The evaluation of the reconstructed time series for the given location focused on: 1) the long-term variability and trends in EUV radiation, 2) the relationships between EUV radiation doses, TOC and cloud cover, and, 3) the days with very high EUV radiation daily doses
and their relationship to large-scale atmospheric circulation patterns.

**2 Study site**

The data used in this study have been obtained at the Solar and Ozone Observatory of the Czech Hydrometeorological Institute, which has been operating since 1951 and focuses mostly on ozone and solar radiation measurements (Vaníček, 2001). The observatory is situated on a small hill in the south of Hradec Králové (50.180° N, 15.833° E) at an altitude of 285
m a.s.l. The building stands away from local pollution sources and the southern horizon is open. During the studied period (1964–2013), the localization of the station was not altered and, except from the change of the instruments, there are no other known sources of data inhomogeneity. All the instruments installed at the Hradec Králové observatory are described in Vaníček (2001) and Vaníček et al. (2015), and have been calibrated regularly according to the standards corresponding to the Guide to Meteorological Instruments and Methods of Observations, World Meteorological Organization (WMO, 2014).

**3 Erythemal UV radiation time series reconstruction**

**3.1 Input data**

The measurements of UV radiation at the Hradec Králové observatory started in 1994 (Vaníček, 2001), so in order to analyze the five-decades long time series, the EUV radiation time series had to be reconstructed. As an input, total ozone column (TOC), atmospheric optical depth (AOD), surface albedo, the water vapor column and global radiation from the
Hradec Králové observatory have been used.

**3.1.1 Total ozone column**

TOC is one of the most important variables affecting the intensities of EUV radiation. The mean daily TOC values for the period 1964–2013 were compiled from the Brewer spectrophotometer MK-IV B098 and the Dobson spectrophotometer D074 measurements. The missing values were estimated from ERA-40 and ERA-Interim reanalysis datasets (Uppala et al.,
2005; Dee et al., 2011). Related instruments and the method used for the mean daily TOC time series completion are described in Vaníček et al. (2012).





### 3.1.2 Aerosol optical depth

Since there are no long-term atmospheric aerosol observations at the study site, annual climatological cycles of AOD were constructed based on the existing records. This method can approximate the climatological behavior of aerosols but no yearly variations are considered (Lindfors et al., 2007). The mean daily AOD values for $\lambda$ = 320 nm ($AOD_{320}$) were

measured by the Brewer spectrophotometer B184 in the period 2005–2013. The mean daily AOD values for $\lambda$ = 550 nm ($AOD_{550}$) for the period 2001–2013 were obtained from the Moderate Resolution Imaging Spectroradiometer (MODIS) (Remer et al., 2005). The completion of the time series, from which the climatological cycle was calculated, was 61 % for $AOD_{320}$, and 46 % for $AOD_{550}$.

### 3.1.3 Surface albedo

Surface albedo is a very important variable determining the intensity of solar UV radiation. In winter months, surface albedo changes may increase the intensity of solar UV radiation by up to 20–30 % (Blumthaler and Ambach, 1988; Krzyścin et al., 2004). The mean daily surface albedo for short-wave radiation in the period 1964–2013 was calculated using an ensemble of multilayer perceptron neural networks with the observed state of the surface as the predictor and the measured short-wave albedo as the predictand. Only the data from noon hours were used, and the possible annual course of the relationship was

taken into account with the help of circular neurons added to the input layer. Data since 2000 were used for the networks training; the trained network was then applied to estimate the albedo in the period 1964–1999, when the measured short-wave albedo was not available. The short-wave albedo was then transferred to the albedo for UV radiation using Eq. (1). The coefficients were estimated based on the short-wave albedo for different states of the surface, because for most surfaces, UV albedo is very low but for snow-covered surfaces it approaches the values of short-wave albedo (Feister and Grewe, 1995;

De Paula Corrêa and Ceballos, 2008).

$$ALB_{UV} = \begin{cases} 0.05 & for\ ALB_{SW} < 0.21 \\ 1.36\ ALB_{SW} - 0.24 & for\ 0.21 \le ALB_{SW} \le 0.65\ , \\ ALB_{SW} & for\ ALB_{SW} > 0.65 \end{cases} \qquad (1)$$

where $ALB_{UV}$ is the albedo for UV radiation and $ALB_{SW}$ is the albedo for short-wave radiation.

### 3.1.4 Water vapor column

Atmospheric water vapor absorbs a significant part of the global solar radiation (e.g. Solomon et al., 1998); therefore, the

total water vapor column was also taken in account when performing the radiative transfer calculations. The daily means of water vapor column in the period 1964–2013 were obtained from the ERA-40 and ERA-Interim reanalysis datasets.





### 3.1.5 Solar radiation

Global solar radiation was used in the EUV radiation time series reconstruction in order to estimate the cloud effects on EUV radiation (e.g. Schwander et al., 2002). In the period 1964–2013, the global radiation was measured using the CM5 and CM11 pyranometers by Kipp & Zonen (Vaníček et al., 2015).

At the Hradec Králové observatory, UV radiation has been measured by the Brewer spectrophotometer MK-IV B098 since 1994. In 2004, the double-monochromator Brewer MK-III spectrophotometer B184 was installed. Using the spectroradiometric method, these instruments are able to detect solar UV radiation in the wavelength interval 280–400 nm (UV-A and UV-B radiation). Both instruments are calibrated regularly every two years against the B017 world traveling standard and the calibration uncertainty is up to 1 % (see e.g. Vaníček, 2002; Vaníček et al., 2015). The EUV radiation time

series obtained by the Brewer spectrophotometer B184 in the period 2005–2013 helped to determine the EUV radiation reconstruction equations and to validate the model (see Sect. 3.2 Methods and model validation).

### 3.2 Methods and model validation

The reconstruction of the 1964–2013 EUV radiation time series from the Hradec Králové observatory was performed using a radiative transfer model and empirical non-linear equations. The time series was originally reconstructed with hourly

resolution, because better results are achieved when a higher time resolution is applied (Rieder et al., 2008). Then, the EUV radiation daily doses were calculated from the reconstructed data.

First, the clear-sky EUV and global radiation were modeled using the libRadtran radiative transfer package (Mayer and Kylling, 2005). The input parameters contained the information on the location (latitude, longitude and altitude), TOC, albedo, annual climatological cycles of atmospheric aerosols ($AOD_{320}$ and $AOD_{550}$), and, in the case of clear-sky global

radiation simulation, total column water.

The all-sky EUV radiation was calculated based on global radiation measurements, which is a common method for reconstructing UV radiation doses (e.g. Krzyścin et al., 2004; Lindfors et al., 2007; Rieder et al., 2008). The reconstruction was based on the cloud modification factor ($CMF_{EUV}$), which is the ratio between cloudy sky EUV radiation ($EUV_{cloud}$) and clear sky EUV radiation ($EUV_{clear}$), as shown in Eq. (2). Correspondingly, the CMF for global radiation ($CMF_{GLB}$) was

calculated as the ratio between cloudy sky ($GLB_{cloud}$) and clear sky global radiation ($GLB_{clear}$), as shown in Eq. (3).

$$CMF_{EUV} = \frac{EUV_{cloud}}{EUV_{clear}} \qquad\qquad (2)$$

$$CMF_{GLB} = \frac{GLB_{cloud}}{GLB_{clear}} \qquad\qquad (3)$$

Therefore, the reconstructed all sky EUV radiation ($EUV_{rec}$) can be expressed as the modeled $EUV_{clear}$ multiplied by $CMF_{EUV}$. $CMF_{EUV}$ was obtained using the global radiation observations, since it can be expressed as the function of $CMF_{GLB}$

and solar zenith angle (SZA) (e.g. Lindfors et al., 2007; Bilbao et al., 2011). To obtain the best-fit function, a multiple non-linear regression was used.





The verification of the model outputs was performed using the individual measurements of EUV radiation from the period 2005–2013. The individual records were coupled with the global radiation data, and the entire data set was then randomly split into two independent data sets with roughly the same number of entries. The first data set was used to develop the model and the second to test it. The time interval between the individual EUV irradiation records was not regular; therefore,

in order to evaluate the model performance, the daily mean EUV radiation was calculated and assessed. Table 1 shows the mean error, the root mean square error and the determination coefficient $r^2$ between the reconstructed and observed mean daily EUV radiation for the whole period and for the seasons (spring = MAM, summer = JJA, autumn = SON, winter = DJF). The errors are lowest in summer and highest in winter, while the correlation between measured and reconstructed daily doses is highest in autumn and lowest in winter, but it exceeds 94 % in all seasons. Fig. 1 shows the absolute and relative

differences between the observed and reconstructed mean daily EUV radiation. Although the absolute differences are small, the relative differences are largest in winter (the mean relative difference makes up to 10 % in February). On the other hand, in spring, summer and autumn, the mean relative differences are in the range of 0–3 %, and less than 20 % of values exceed the 10 % difference between the observed and reconstructed daily means.

**4 Methods of the time series analysis**

The analysis of the reconstructed 1964–2013 EUV radiation consists of three parts: 1) the annual variations and trends, 2) the factors affecting EUV radiation, and 3) the analysis of days with very high EUV radiation doses. The data were processed using the software STATISTICA®10, R Project for Statistical Computing (R core team, 2014), and ArcGIS®10.

To perform the analyses, TOC and surface albedo data (Sect. 3.1) were used. Hourly cloud cover data in octas were included in the analysis in order to evaluate the cloud effects on EUV radiation. There are no cloud cover records directly at the

Hradec Králové observatory, therefore, the data from the Hradec Králové, Pouchov synoptic station (50.246° N, 15.843° E; 243 m a.s.l.; 8 km north of the Hradec Králové observatory) were used. The missing values in the time series were supplemented with data from the Pardubice, airport synoptic station (located in flat lowlands at 50.013° N, 150739° E; 225 m a.s.l.; 20 km south of the Hradec Králové observatory). The linear regression relationship between these two datasets was statistically significant ($r^2$ = 73 %). For every day, the weighted mean of cloud cover was calculated, while the weight was

set as the mean clear-sky EUV radiation at the certain time of the day. The gaps in the aggregated daily cloud cover time series (7 % of days) were then filled by a non-linear regression using the global radiation daily doses from the Hradec Králové observatory ($r^2$ = 76 %).

The trends in EUV radiation, as well as TOC and cloudiness, were studied using locally weighted scatterplot smoothing (Cleveland, 1979). The current trends in TOC were evaluated using linear regression in the period 1995–2013, which was

selected for this assessment because 1995 was the year when the long-term ozone lowering over central Europe stopped (Krzyścin and Borkowski, 2008). The effect of SZA, TOC and cloud cover on EUV radiation daily doses and their monthly



and yearly means throughout the entire study period and the five decades was analyzed using partial correlation coefficients ($r_{part}$). The level of statistical significance α was set to 0.05 in all the tests performed in this study.

The emphasis was placed on days with very high EUV radiation doses (days with $EUV_{90+}$). They were defined based on the 90th percentile of the EUV radiation daily doses for each month. Therefore, 1827 days with $EUV_{90+}$ were selected and further

analyzed. The monthly threshold values of $EUV_{90+}$ for the period 1964–2013 are displayed in Table 2. The days with $EUV_{90+}$ were then assessed with respect to trends and explained with regards to TOC, cloud cover and albedo. Low TOC was defined as the 50th percentile of each month and lower. According to Alados-Arboledas et al. (2003), high intensities of EUV radiation can be observed under the cloud cover or 4 octas or less, therefore, low cloud cover was specified this way. High albedo was set to 0.3, which is, according to De Paula Corrêa and Ceballos (2008), the lowest albedo of consistent snow

cover. The pressure field in the days with $EUV_{90+}$ was investigated using principal component analysis (Storch and Zwiers, 1999). Two geopotential heights were chosen: 1000 hPa, which is well-representative of surface pressure, and 70 hPa, which is the level near the ozone layer maximum where the ozone changes were most evident (Kirchner and Peters, 2003). The field of geopotential heights for pressure levels 1000 and 70 hPa was obtained from the NCEP/NCAR reanalysis (Kalnay et al., 1996). The study area was defined as a rectangle between 40° W – 40° E and 20° N – 70° N with a spatial resolution of

2.5°. The daily values of the principal components were then correlated with the Northern Atlantic Oscillation (NAO) and Arctic Oscillation (AO) indices (Wallace and Gutzler, 1981; Thompson and Wallace, 1998).

## 5 Results and discussion

### 5.1 Annual variations and trends

During the period 1964–2013, the annual means of TOC (Fig. 2a) show a large relative variability (coefficient of variation $c_v$

= 3.5 %) with diverse trends in the individual decades (Table 3). In the 1960s and 1970s, the annual mean TOC at the Hradec Králové observatory fluctuated between 340 and 360 DU. The decline in TOC started at the beginning of the 1980s, reaching the minimum in 1993 following the eruption of Mount Pinatubo (annual mean of 315 DU). Then the TOC fluctuated at lower values than at the beginning of the studied period (320–350 DU). The most distinct decline in TOC was recorded in spring; the weakest decline was observed in autumn (Fig. 2d). Since 1995, the linear trends of TOC were not statistically

significant in any particular month, but there was an increase in winter (January: 1.2±0.7 DU per year, February: 1.3±0.9 DU per year) and a decrease in summer (August: -0.4±0.4 DU per year). The trends in TOC, which are linked mostly to the increase in ozone depleting substances in the stratosphere, but also to the changes of atmospheric dynamics, show a very similar development to other parts of central Europe (e.g. Trepte and Winkler, 2004; Hood and Soukharev, 2005; Harris et al., 2008). The negative trends in warm seasons after 1995, which can be linked to the changes of circulation patterns

affecting the concentration of ozone depleting substances in the atmosphere, were also recorded at other parts of central Europe (Krzyścin and Borkowski, 2008; Vaníček et al. 2012; Krzyścin and Rajewska-Więch, 2015). The annual variation of TOC recorded at the Hradec Králové observatory, with its minima in autumn (October) and maxima in spring (March or





April), is connected to the natural variability of TOC in Europe, which is linked to the transport of ozone from low latitudes (Zvyagintsev et al., 2015).

Annual cloud cover means (Fig. 2b) exhibit a larger relative variability than TOC ($c_v$ = 5.3 %) and changeable trends (Table 3). Cloud cover maxima were recorded in winter (November or December) and minima in summer (August), while in some

years there was a secondary maximum present in June or July (attributed to prevailing convection). Over the period 1964–2013, cloud cover did not show any compact trend in any of the months (Fig. 2e). There was a period of high cloud cover during the end of the 1970s and the beginning of the 1980s, which was most pronounced in July and October, with the maximum yearly mean cloud cover in 1981 (6.0 octas). This period was followed by a decrease in cloud cover in the 1990s, which was very distinct in January, May, July, and October. Since the 1990s, the yearly mean cloud cover increased up to

5.9 octas in 2013. According to Wibig (2008), who observed similar patterns in Poland, the most important factor determining the changes in cloud cover is the variability of the occurrence of different cloud types, for example the intensification of convection in summer months. The changes in the 1990s seem to be driven especially by the decreasing amount of sulfur dioxide emitted into the atmosphere (e.g. Krüger and Graßl, 2002).

Of the three studied variables, the annual mean daily doses of EUV radiation (Fig. 2c) show the largest relative variability ($c_v$

= 7.4 %). The trends in the annual mean daily doses of EUV radiation were also changing over the study period (Table 3). The EUV radiation doses declined until the end of the 1970s, which might indicate a connection to the increase of cloudiness in this period. In the 1980s and 1990s, the EUV doses increased steeply due to the decrease in TOC, with the maximal mean daily dose of EUV radiation being recorded in 2003. The slight decline in EUV radiation doses since about 2005 was mostly attributed to the changes in cloud cover. The relative increase in EUV radiation doses was most pronounced in spring and

summer, less in autumn and winter. The overall changes in EUV radiation doses were not statistically significant in January, September, October, or December. These results are in accord with most other publications focusing on the long-term variability of EUV radiation in central Europe (e.g. Rieder et al., 2008; Den Outer et al., 2010; Krzyścin et al., 2011). The very high EUV radiation doses in 2003, which were caused by the anomalously low cloud cover due to high pressure episodes in summer, were also recorded at other European stations (Den Outer et al., 2010; Rieder et al., 2010). The annual

variations of EUV radiation doses, shown in Fig. 2f, are clearly following the changes in SZA with maxima in summer (June, July) and minima in winter (December). The shift of the annual maxima from June to July can be attributed to the annual variations of TOC (Seckmeyer et al., 2008).

### 5.2 Factors affecting the erythemal UV radiation doses

The importance of the factors affecting the EUV radiation doses varied based on the chosen time period, i.e. daily doses and

their monthly and annual means. Over the period 1964–2013, the dominant factor affecting the annual means of EUV radiation daily doses was TOC (partial correlation coefficient $r_{part}$ = -0.78), yet the effect of cloud cover was also statistically significant ($r_{part}$ = -0.46). The monthly means of EUV radiation daily doses were affected mostly by cloud cover, the effect of TOC was significant in all months except January, February and December; in June and August it was greater than the effect





of cloud cover (Table 4). The daily doses of EUV radiation were most affected by SZA, cloud cover, and then by TOC ($r_{part}$ = -0.88, -0.58, and -0.30, respectively). During the individual months, cloud cover was the most important factor affecting the EUV radiation daily doses; TOC had the most pronounced effect in April. Furthermore, the effect of SZA, TOC, and cloud cover varied throughout the study period. The effect of SZA did not change much over the decades; it ranged from $r_{part}$

= -0.87 to $r_{part}$ = -0.89. It was the strongest in the months when SZA is the most variable (spring and autumn months), but it had insignificant effect in the months near the solstices (Fig. 3a). The effect of TOC was most pronounced in the decade 1984–1993 ($r_{part}$ = -0.31), when the thinning of the ozone layer was most significant. Figure 3b shows TOC had the strongest effect on the daily EUV radiation doses in the spring months, but also in late summer and early autumn, especially in the decades 1984–1993 and 1994–2003. The effect of cloud cover was the strongest in the decade 1994–2003 ($r_{part}$ = -0.62),

while the weakest relationship was found in the decade 2004–2013 ($r_{part}$ = -0.56). As shown in Fig. 3c, the effect of cloud cover also differed in individual seasons, with the strongest influence in the summer months of the decades 1964–1973, 1984–1993 and 2004–2013, and in the spring and autumn months of the decade 1974–1983.

The results concur with similar studies performed at other European locations, showing that TOC is the main driver of long-term EUV radiation changes, while SZA and cloud cover are the dominant factors affecting EUV radiation in the short-term

(Den Outer et al., 2010; Herman, 2010; De Bock et al., 2014). The changes in EUV radiation in the 1970s have been attributed to cloud cover variability, which is in accord with the high $r_{part}$ shown in Fig. 3c. The latter increase in EUV radiation doses was caused mainly by the strong total ozone loss. For example, in the late 1980s to early 2000s, the changes in TOC were responsible for the majority of changes in EUV radiation in Austria, especially in spring (Rieder et al., 2008). At the beginning of the new millennium, the effect of cloud cover grew stronger in contrast to the previous decades

(Krzyścin et al., 2004).

### 5.3 Very high erythemal UV radiation daily doses

The EUV radiation doses reconstructed for the Hradec Králové observatory within the chosen study period have been examined with regards to very high values ($EUV_{90+}$). Based on the 90th percentile of each month (see Sect. 4 and Table 2), a total of 1827 days with $EUV_{90+}$ were selected for further analysis. In the 1960s and the beginning of the 1970s, the number

of days with $EUV_{90+}$ decreased, followed by a steep increase, which stopped at the end of the 1990s. The number of days with $EUV_{90+}$ increased again at the beginning of the 21st century. The general increase in days with $EUV_{90+}$ was statistically significant in all seasons except autumn, while it was most evident in spring (an increase by 3 days in 10 years, $r^2$ = 47 %). The factors affecting the occurrence of days with $EUV_{90+}$ in all the studied decades are shown in Fig. 4. There was a statistically significant negative correlation of the yearly number of days with $EUV_{90+}$ and mean yearly TOC ($r^2$ = 61 %) and

with mean yearly cloud cover ($r^2$ = 23 %). The effect of TOC was strongest in spring ($r^2$ = 53 %) and weakest, but still statistically significant, in winter ($r^2$ = 8 %). Cloud cover showed the most significant effect in summer ($r^2$ = 17 %) and its effect was insignificant in autumn. 82 % of days with $EUV_{90+}$ were recorded when the TOC was low (less than the 50th percentile of each month), but in spring and summer this value exceeded 94 %. Moreover, 74 % of days with $EUV_{90+}$





occurred when the daily mean cloud cover was 4 octas or lower; in summer this value increased to 87 %. High albedo was observed in 19 % of the days with $EUV_{90+}$, in winter in 74 % of days with $EUV_{90+}$. Most of the unexplained days with $EUV_{90+}$ occurred in autumn, when the studied factors were also less likely to combine. Therefore, the yearly number of days with EUV90+ is both in total and in all seasons affected more by TOC than by cloud cover. The increase in days with high

EUV radiation doses was also observed in Austria and Switzerland, especially in the 1990s. Low TOC and its combination with partly cloudy or cloudless skies were also the most frequent causes of days with high EUV radiation doses. (Rieder et al., 2010).

The relationship between the very high EUV radiation doses and the atmospheric circulation was studied using the principal component analysis (PCA) and the Arctic Oscillation (AO) and North Atlantic Oscillation (NAO) indices. The first and

second PCA modes (Fig. 5) show that the occurrence of days with $EUV_{90+}$ is linked to NAO, AO, and the shape and strength of the Arctic circumpolar vortex. The 1000 hPa first component clearly indicates the effect of the Azores High promontory in central Europe. The intensities of the Azores High and Icelandic Low are statistically interconnected (Wallace and Gutzler, 1981); therefore, both these pressure systems are linked to the occurrence of days with $EUV_{90+}$. This can be supported by the statistically significant negative correlation coefficients between the 1000 hPa first PCA component and the

AO and NAO indices (Table 5). Although the relationship between air pressure systems and shortwave radiation is expressed mostly through the redistribution of clouds (Chiacchio and Wild, 2010), the results presented in this study indicate that low TOC is the dominant factor affecting the incidence of days with $EUV_{90+}$. Low TOC occurs mostly during the NAO positive phase, which leads to stationary planetary waves, stronger zonal winds, and therefore a colder Arctic circumpolar vortex (Orsolini and Limpasuvan, 2001; Schnadt and Dameris, 2003). A cold polar vortex allows the development of polar

stratospheric clouds, which leads to greater ozone depletion over the Northern hemisphere (Harris et al., 2010). Moreover, during the positive NAO phase, a ridge of high air pressure develops over central Europe (Hurrell, 1995), which usually leads to little cloudiness that enables a more frequent occurrence of days with $EUV_{90+}$. The second 1000 hPa PCA mode can be associated with the spatial pattern of high- and low-air pressure areas south of Iceland and over Scandinavia. It can therefore be understood as the meridional airflow over the North Sea, either from or to the area of the Arctic circumpolar

vortex.

The zonal character of the 70 hPa first PCA mode in spring, autumn, and winter (Fig. 5) is linked to the strength of the Artic circumpolar vortex. It seems to be the dominant stratospheric factor affecting the amount of ozone and therefore also the occurrence of days with $EUV_{90+}$ in these seasons. The Arctic circumpolar vortex decays in summer (Harvey et al., 2002), so the polar front cyclones can easily penetrate it, upset its symmetry and affect the transport of ozone from the lower latitudes.

Subsequently, this situation also favors the occurrence of days with $EUV_{90+}$, which is especially affected by the influx of air from the south-west. The first PCA mode in summer and the second PCA mode in other seasons document this meridional component of atmospheric circulation, which allows ozone-poor air from the tropics to flow into central Europe as well as high-latitude regions. Moreover, these conditions are often accompanied by high air pressure leading to clear-sky or partly cloudy situations (Stick et al., 2006), thus having major influence on the incidence of high EUV radiation doses.





### 6 Concluding remarks

In this paper, the reconstruction and analysis of the EUV radiation daily dose time series from the Solar and Ozone Observatory in Hradec Králové (Czech Republic) have been performed for the period 1964–2013. The time series was reconstructed using a radiative transfer model and an empirical relationship of the observed EUV and global solar radiation.

The model was verified based on EUV radiation observations of the Brewer spectrophotometer B184 in the period 2005–2013. The selected methods applied for the time series reconstruction gave the best estimates of EUV radiation using the input and validation data (solar radiation, TOC, albedo, AOD, water vapor column) available for the given location. This approach significantly improved the quality and accuracy of the reconstructed EUV radiation time series, which was extended to the entire 1964–2013 period.

The study focused on the evaluation of general variations and long-term trends, and the factors affecting EUV radiation doses. The results showed that the EUV radiation daily doses were characterized by large variability and changeable trends in the individual decades. Due to increasing cloud cover and TOC fluctuations, the EUV radiation doses slightly decreased in the 1960s and 1970s. The rapid increase in EUV radiation daily doses, observed in the 1980s and 1990s, was linked to the TOC decline, supporting the findings of previous studies on TOC variation and past EUV radiation doses. At the beginning

of the new millennium, EUV radiation fluctuations were mostly attributed to the changes in cloud cover. Moreover, the results confirmed that the annual means of EUV radiation daily doses are most affected by changes in TOC, while the monthly means are most influenced by cloud cover. The daily EUV radiation doses were mainly affected by SZA, but the effect of SZA was insignificant in the summer and winter solstice months.

The number of days with very high EUV radiation doses (EUV$_{90+}$) increased significantly throughout the study period,
especially in the spring months. The occurrence of days with EUV$_{90+}$ was mostly affected by low TOC, but clear or partly cloudy skies also had a significant role, especially in summer. In winter months, the increased surface albedo also significantly affected the occurrence of days with EUV$_{90+}$. Therefore, it was clearly indicated that the main factors affecting the long-term changes of the occurrence of days with EUV$_{90+}$ are seasonally dependent. For the first time, the relationship between the high EUV radiation doses and synoptic weather situations was evaluated using principal component analysis
and large-scale atmospheric circulation patterns at the 1000 hPa and 70 hPa geopotential heights. The results suggest that days with EUV$_{90+}$ occur most likely during the positive NAO phase, when the Azores High promontory reaches over the area of central Europe. These conditions lead to a cold and stable Arctic circumpolar vortex, which contributes to accelerated ozone depletion over the Arctic region. Moreover, a high air pressure ridge over central Europe frequently causes cloudless skies and subsequently higher EUV radiation doses. In summer, the influx of ozone-poor air from the south-west in the
upper levels of the atmosphere can also contribute to the occurrence of days with EUV$_{90+}$.

Compared to the existing long-term datasets, the reconstructed EUV radiation time series is one of the longest in central Europe, and therefore can be used for the investigation of the long-term photobiological effects on organisms, or for skin cancer research. Furthermore, the study brought valuable knowledge of regional EUV radiation variability and trends in the





past decades. It especially provided a closer look at the days with very high EUV radiation doses and their relationships with other variables, including the atmospheric circulation patterns. Nevertheless, the EUV radiation doses and the trends described in this study might be significantly affected by ozone layer recovery and by the changes of atmospheric circulation patterns predicted by the atmospheric chemistry and climate models (e.g. Inglesias-Suarez et al., 2016). It must, however, be

anticipated that the scope of the study has not allowed the taking into account of other possibly important factors, such as the El Niño Southern Oscillation or Quasi-biennial Oscillation, which will be considered in further research.

**Data availability**

The data used for the study are property of the Czech Hydrometeorological Institute, Hradec Králové, Czech Republic and are the subject of the data policy of the above-mentioned institution. Any person interested in the underlying data should

contact Ladislav Metelka, the head of the Solar and Ozone Observatory of the Czech Hydrometeorological Institute, Hradec Králové (email: metelka@chmi.cz).

**Author contribution**

K. Láska, L. Metelka and K. Čížková drew up the research idea, K.Čížková and L. Metelka prepared the data, M. Staněk wrote the libRadtran script and K. Čížková carried out the time series reconstruction and the analyses. K. Čížková and K.

Láska prepared the manuscript with contributions from L. Metelka.

**Competing interests**

The authors declare that they have no conflict of interest.

**Special issue statement**

The work was presented at the Quadrennial Ozone Symposium 2016 in Edinburgh (abstract No. QOS2016-57) and we have

expressed the interest to submit the paper to the joint Special Issue "Quadrennial Ozone Symposium 2016 – Status and trends of atmospheric ozone" in ACP/AMT journals following on the QOS2016.

**Acknowledgments**

The research was supported by the project of the Czech Hydrometeorological Institute No. 03461022 'Monitoring of the ozone layer and UV radiation in Antarctica', which is funded by the State Environmental Fund of the Czech Republic and by

the project of Masaryk University MUNI/A/1419/2016. The work was also supported by the ECOPOLARIS project No.



CZ.02.1.01/0.0/0.0/16_013/0001708 for the co-operation on data analysis. Data on total ozone content and water vapor column have been taken from the ERA-40 and ERA-Interim reanalysis issued by the European Centre for Medium-Range Weather Forecasts; the aerosol optical depth data have been provided by the NASA Goddard Earth Sciences Data and Information Services Center (at http://giovanni.gsfc.nasa.gov), and the geopotential height fields were acquired from the NCEP/NCAR reanalysis of the Earth System Research Laboratory, National Oceanic and Atmospheric Administration (https://www.esrl.noaa.gov/psd/data/reanalysis/reanalysis.shtml).

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





**Table 1. Mean Error (ME) and its percentage value (ME%), Root Mean Square Error (RMSE) and its percentage value (RMSE%), and the determination coefficient (r$^2$) for the reconstructed time series and for the individual seasons at the Hradec Králové observatory for 1964–2013.**

| Season | ME | ME% | RMSE | RMSE% | r$^2$ (%) |
|---|---|---|---|---|---|
| Year | 0.05 | 1.01 | 2.47 | 9.93 | 99.1 |
| Spring | 0.44 | 1.43 | 2.93 | 8.28 | 98.2 |
| Summer | -0.17 | -0.46 | 3.21 | 5.66 | 98.0 |
| Autumn | -0.60 | -2.12 | 1.67 | 7.85 | 99.2 |
| Winter | 0.58 | 5.61 | 1.29 | 15.69 | 94.6 |



**Table 2. Threshold values of EUV radiation daily doses for days with EUV90+ in individual months.**

| Month | I | II | III | IV | V | VI |
|---|---|---|---|---|---|---|
| $EUV_{90+}$ (kJ m$^{-2}$) | 0.30 | 0.65 | 1.31 | 2.39 | 3.36 | 3.97 |
| Month | VII | VIII | IX | X | XI | XII |
| $EUV_{90+}$ (kJ m$^{-2}$) | 3.92 | 3.22 | 2.09 | 1.08 | 0.38 | 0.26 |





**Table 3. Linear trends and standard error of annual means of TOC, cloud cover and EUV radiation doses for the individual decades of the period 1964–2013 at the Hradec Králové observatory; stars mark statistically significant trends (α = 0.05)**

| Decade | TOC (% per 10 yr.) | Cloud cover (% per 10 yr.) | EUV doses (% per 10 yr.) |
|---|---|---|---|
| 1964–1973 | 4.3 ± 2.0 | -1.2 ± 5.8 | -3.0 ± 6.0 |
| 1974–1983 | 0.2 ± 2.7 | -5.7 ± 7.1 | 7.7 ± 6.0 |
| 1984–1993 | -9.4 ± 2.4* | -11.1 ± 4.1* | 14.9 ± 6.1* |
| 1994–2003 | 2.31 ± 1.9 | 0.6 ± 6.3 | 2.5 ± 5.4 |
| 2004–2013 | -1.0 ± 2.9 | 3.1 ± 5.2 | -0.5 ± 4.7 |





**Table 4. Partial correlation coefficients of the effect of TOC and cloud cover on the monthly means of EUV radiation daily doses throughout the period 1964–2013; stars mark statistically significant correlations ($\alpha = 0.05$).**

| Period | SZA | TOC | Cloud cover |
|---|---|---|---|
| Year | -0.88* | -0.30* | -0.58* |
| I | -0.55* | -0.15* | -0.70* |
| II | -0.61* | -0.29* | -0.70* |
| III | -0.63* | -0.39* | -0.76* |
| IV | -0.60* | -0.51* | -0.79* |
| V | -0.35* | -0.49* | -0.80* |
| VI | -0.05 | -0.43* | -0.80* |
| VII | -0.29* | -0.43* | -0.81* |
| VIII | -0.58* | -0.46* | -0.79* |
| IX | -0.69* | -0.45* | -0.79* |
| X | -0.77* | -0.36* | -0.79* |
| XI | -0.56* | -0.21* | -0.75* |
| XII | -0.16* | -0.10* | -0.69* |



**Table 5. The correlation coefficients between the 1000 hPa first PCA component and the NAO and AO indices; stars mark statistically significant correlations (α = 0.05).**

| Season | NAO | AO |
|--------|-------|-------|
| Spring | -0.54* | -0.59* |
| Summer | -0.43 | -0.51* |
| Autumn | -0.35 | -0.51* |
| Winter | -0.63* | -0.68* |


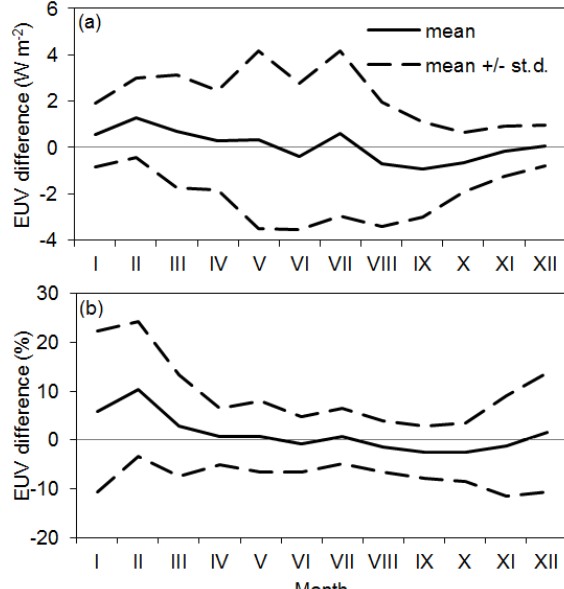

**Figure 1.** The mean (a) absolute and (b) relative differences between the observed and modeled EUV radiation intensities and their standard deviations (st.d.) at the Hradec Králové observatory in the period 2005–2013.





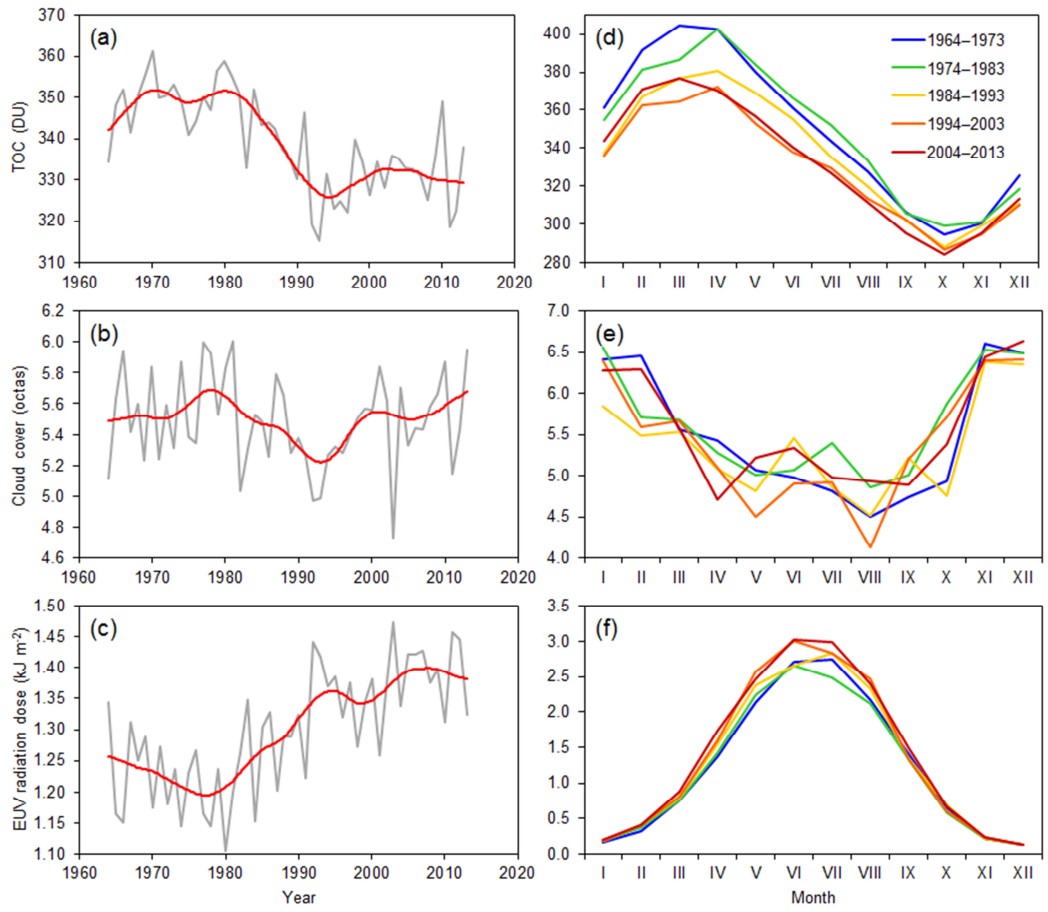

**Figure 2.** The annual means of (a) TOC, (b) cloud cover, and (c) daily EUV radiation doses fitted with the locally weighted scatterplot smoothing curve (red); and the mean annual variation of (d) TOC, (e) cloud cover, and (f) daily EUV radiation doses at the Hradec Králové observatory in the individual decades of the period 1964–2013.



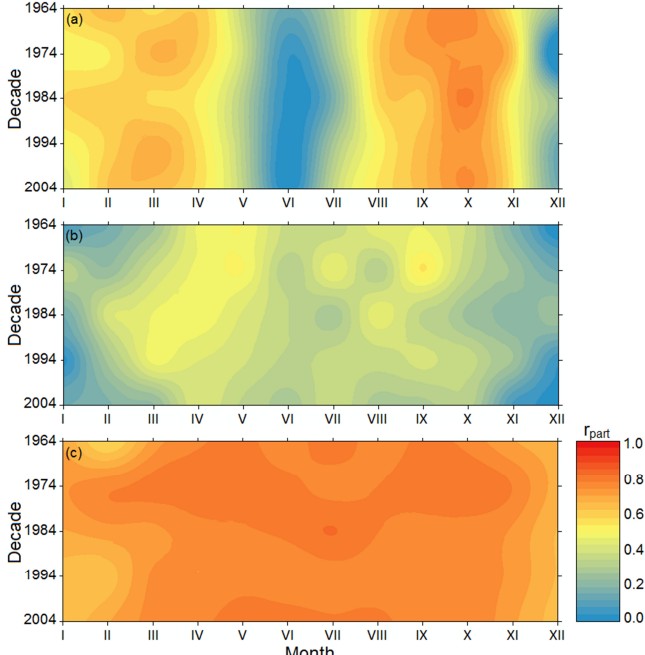

**Figure 3. Partial correlation coefficients ($r_{part}$) of the effects of (a) SZA, (b) TOC, and (c) cloud cover on the daily doses in individual months and decades; the decades are labeled only by their initial years.**



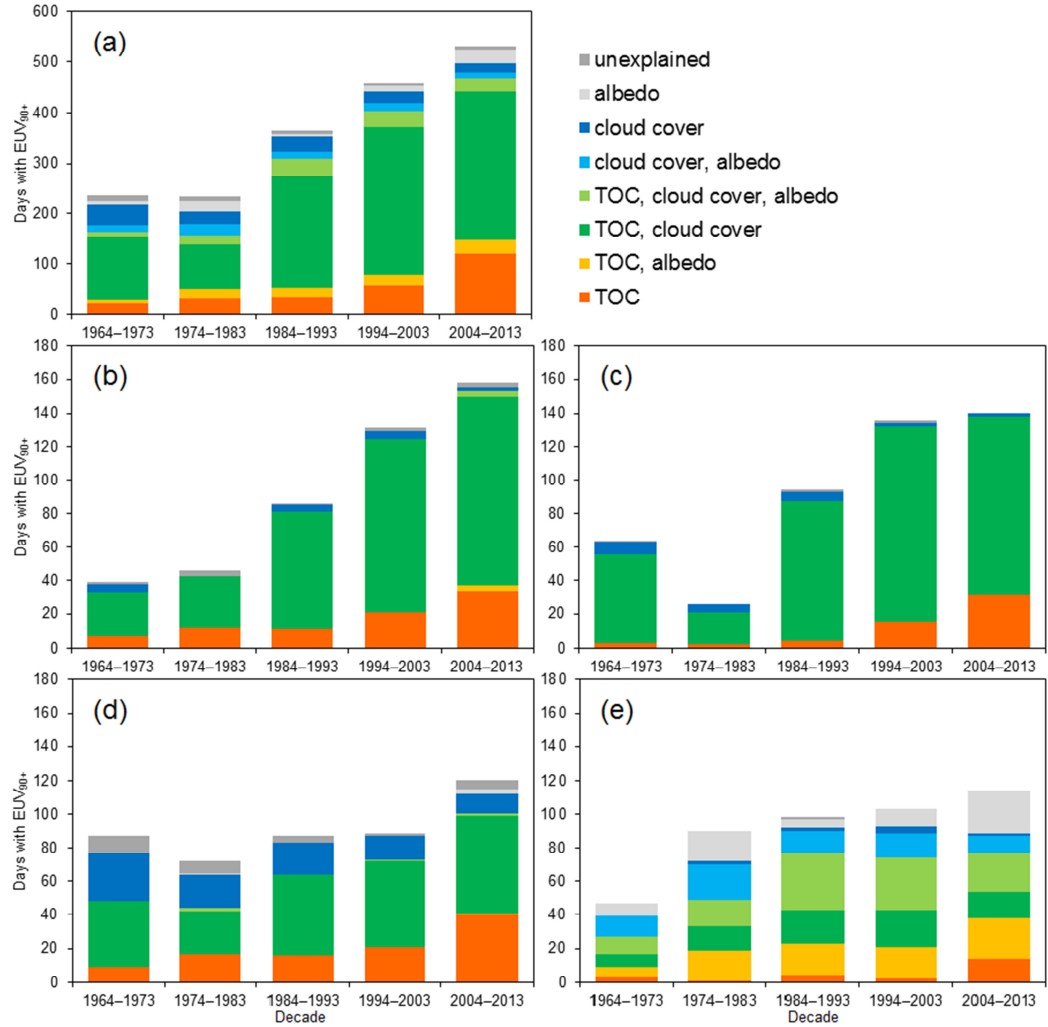

**Figure 4. The number of days with EUV$_{90+}$ in the individual studied decades, and their explanation by TOC, cloud cover, albedo and their combinations, for (a) the entire year, (b) spring, (c) summer, (d) autumn, and (e) winter.**




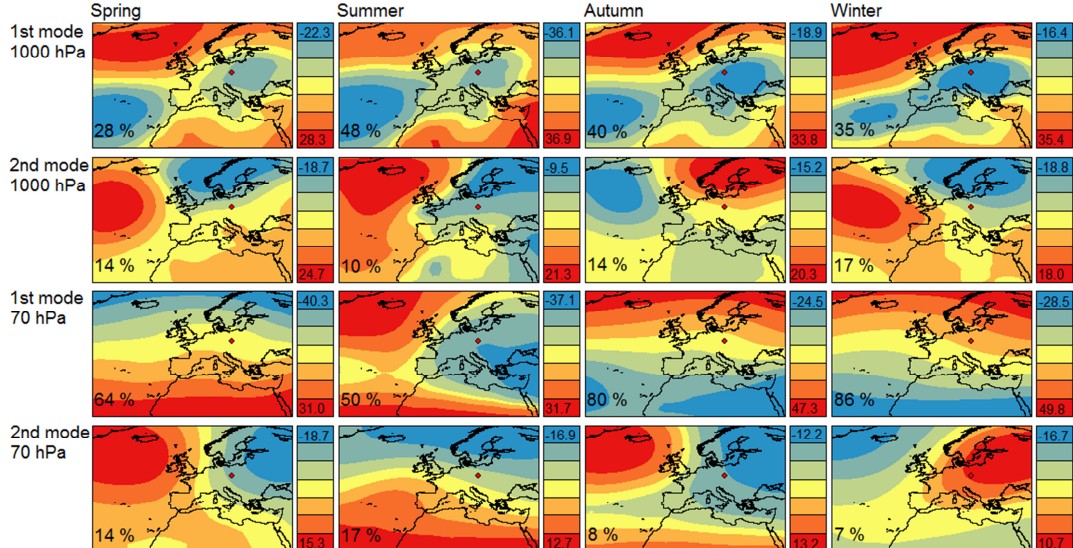

**Figure 5.** The values of the first and second PCA modes of the 1000 hPa and 70 hPa geopotential heights for the days with $EUV_{90+}$ in spring, summer, autumn, and winter (represented by the color legend on the right). Variability explained by each of the modes is indicated by the relative value in the bottom-left corner. The red dot marks the location of the Hradec Králové observatory.