# Peer review of "Reconstruction and analysis of erythemal UV radiation time series from Hradec Králové (Czech Republic) over the past 50 years"

_Atmospheric Chemistry and Physics, 2017_

## Referee Comment (RC1) · Anonymous Referee #1 · 28 Sep 2017

The paper "Reconstruction and analysis of erythemal UV radiation time series from Hradec Králové (Czech Republic) over the past 50 years" partly reconstructs and analyzes a 50-year time series of UV Erythemal UV radiation in Central Europe. The paper is well structured and state-of-the-art methods are used both for the reconstruction and for the analysis of the time series. The paper also includes innovative aspects, especially the relationship of high EUV daily doses and macro-scale circulation patterns is to the best of my knowledge unique.

Therefore I suggest the acceptance of the manuscript in ACP with minor revisions.

remarks:

[Figure]

Page 2 Line 6 and caption figure 1:

suggestions for changes

radiation intensity –> irradiance

Why do you show mean daily irradiance [W/m$^2$] in figure 1? Daily dose [KJ/m$^2$] would be more consistent with the remainder of the paper.

Page 4 Line 15-17:

How long was the training data set? Could you comment on the relationship of the observed short-wave radiation albedo with values one would expect according to land cover and vegetation cycle?

Page 5 Line 17-20:

Which radiative transfer model did you use (DISORT, TWOSTREAM, MYSTIC, ...)? How do you deal with SZA values close to 90°?

Page 5 Line 30/31:

Can you describe the determination of CMFEUV as f(CMFGLB,SZA) in more detail? Did you generate the cloud modification table according to Lindfors et al. 2007 ?

Figure 5:

It would help the reader, if you would use the same scale for each subfigure (e.g. [-50,50]) both to save space (just one color bar needed) and to enable comparison between subfigures. The effect of SZA on radiation daily doses is trivial. I suggest to skip the analysis (table 4, second column and figure 3a) or to investigate the influence of surface albedo instead.

Typos Page 7, line 8 "cover or" –> cover of

―――――――――――――――――――――――

2017.

---

## Referee Comment (RC2) · Anonymous Referee #2 · 3 Oct 2017

General comments

The paper discusses the reconstruction and analysis of 50-year time series of Erythemal UV radiation (UV-Ery) over the central European station Hradec Králové. The paper also addresses the connection of UV-Ery long-term changes (examining also the evolution of high UV-Ery doses) to large circulation patterns over Europe. The statistical methods applied are well suited for this type of analysis, and the paper contains new material.

However, the paper needs minor improvements before accepted for publication.

The way the statistical methods are applied and the quantities used are not clearly

described, and in some paragraphs may be even confusing. For example, in Page 1, line 18 (Abstract) it is written that "the number of days with very high EUV radiation increased by 22% per decade", a statement that cannot be easily deduced by either the description of methods (paragraph 4) or the discussion in the relevant paragraph (5.3)

Specific comments:

Page 3, lines 24-26: Is it possible to give an indication of the number of missing obs

Page 4, lines 7-8 (3.1.2 AOD): The meaning of this last sentence is not clear. Do you mean that the number of data available for the calculation of the AOD320 climatology was 61% of the total?

Page 5, lines 25 – 30: Please give more details. Is there a reference you can give here? last line (30-31): "To obtain the best fit function.." Where is this best-fit function used?

Page 6 line 1: Please change "The verification.." to "The validation.." lines 3-4: "used to develop the model and the second ..." Which model are you referring to here? LibRadtran? Please clarify. line 30: "because 1995 was the year when the long-term ozone lowering stopped" It is better to write "because stratospheric (or total) ozone reached its minimum in 1995 over Europe"

Page 7 line 8:typo "or" -> "of" line 8: "..therefore, low cloud cover was specified this way". Do you mean that the threshold for low cloud cover was set to 4 octas or less? lines 10-15: Please give the period you used from the NCEP reanalysis line 15: Why do you use both indices? They are not independent from each other. See also comment below

Page 8-9, Par. 5.2 and in connection to Figure 3. Please correct or discuss why this is done this way the figure. Are the correlation coefficients you found and discuss negative or positive? The text says negative, while the figure suggests otherwise.

Page 9, paragraph 5.3: There is no clear description of the methods you used here, nor a clear description of what Figure 4 presents and how it was calculated. Here you discuss only days with high erythemal dose, not all available days. It seems that this is a result of partial correlation performed separately on the EUV90 or how was it done? How was the number referred to in the Abstract (22% per decade increase in the number of days) calculated?

Page 10, lines 8 and below: Why do you use separately NAO and AO for your table 5 correlations? Do you have an explanation as to their individual effects and why they should be examined separately? They are very closely connected, and there is no need to present both. It is better to discuss the effects as joined, as you do in the rest of the paragraph with the PCA analysis.

---

## Author Comment (AC1) · 30 Nov 2017

Dear referees,

Thank you very much for your comments. Enclosed please find the answers to them (referee1_answers.pdf and referee2_answers.pdf), and the manuscript with all the changes (ACP_manuscript_changes.pdf). The changes in the manuscript that are relevant to the 1st referee's comments are marked by yellow highlights, the changes relevant to the 2nd referee's comments are marked by green highlights.

On behalf of all co-authors, Yours faithfully, Klara Cizkova

[Figure]

Please also note the supplement to this comment:
https://www.atmos-chem-phys-discuss.net/acp-2017-701/acp-2017-701-AC1-
supplement.zip

---

## Author Response (AR1)

The paper "Reconstruction and analysis of erythemal UV radiation time series from Hradec Králové (Czech Republic) over the past 50 years" partly reconstructs and analyzes a 50-year time series of UV Erythemal UV radiation in Central Europe. The paper is well structured and state-of-the-art methods are used both for the reconstruction and for the analysis of the time series. The paper also includes innovative aspects, especially the relationship of high EUV daily doses and macro-scale circulation patterns is to the best of my knowledge unique. Therefore I suggest the acceptance of the manuscript in ACP with minor revisions.

Remarks:

1.  Page 2 Line 6 and caption figure 1: Suggestions for changes radiation intensity –> irradiance

    *Changed accordingly.*

2.  Why do you show mean daily irradiance [W/m2] in figure 1? Daily dose [KJ/m2] would be more consistent with the remainder of the paper.

    *The reason for the use of irradiances and not daily doses for the model validation was that the Brewer spectrophotometer EUV radiation measurements, which were used to validate the model, were taken at irregular time intervals, often only several times a day. Therefore, the daily doses calculated from the Brewer data would be very imprecise.*
    *After the validation of the model, we calculated the irradiance for every hour at a given day over the whole 50-years period, therefore an integration to daily doses was possible for the rest of the study.*
    *Clarified in the text (p. 6, l. 8 – 11).*

3.  Page 4 Line 15-17: How long was the training data set? Could you comment on the relationship of the observed short-wave radiation albedo with values one would expect according to land cover and vegetation cycle?

    *The whole dataset comprised the data from the 2000–2014 period (4269 applicable cases). For network training, these cases were randomly divided to training set (70 % of cases), testing set (15 %), and validation set (15 %). An ensemble of 10 individual neural networks was trained, with different random divisions to training/testing/validation subsets. The final results were taken from the ensemble average. Sensitivity tests indicated the dominant role of land cover as a single relevant predictor. From the land cover characteristics, the snow cover was the most important one (land without snow, partly covered by snow or full snow cover).*
    *Clarified in the text (p. 4, l. 15–17).*

4.  Page 5 Line 17-20: Which radiative transfer model did you use (DISORT, TWOSTREAM, MYSTIC,...)? How do you deal with SZA values close to 90◦?

    *We used the DISORT solver (added to the text of the manuscript at p. 5, l. 17).*

*The values close to 90° were calculated by the model, however, they weren't added to the training dataset because of the lower precision of both the model and the instruments (we decided to exclude all values exceeding 75° of SZA). The final model for all-sky conditions was then tested on a dataset that included all values including those with SZA close to 90°. The absolute error was very small, i.e. up to 2.5 W·m⁻² (see fig. res-1 for illustration, and fig. 1 in the manuscript), so it did not have a large effect on the resulting daily doses.*
*Clarified in the text of the paper (p. 6, l. 6–8).*

[Figure]

*Fig. res-1. An example of observed and modeled irradiances for three different days: (a) clear summer day, (b) partly cloudy summer day, and (c) partly cloudy winter day*

5.  Page 5 Line 30/31: Can you describe the determination of CMFEUV as f(CMFGLB,SZA) in more detail? Did you generate the cloud modification table according to Lindfors et al. 2007?

*In order to minimize the RMSE, we did not create the cloud modification look-up table. Instead, we have decided to go for a multiple non-linear regression instead. Here is the formula and the coefficients we used:*

$$CMF_{EUV} = \left[ (a + b\ SZA)\ CMF_{GLB}^{(c+d\ SZA)} \right] \cdot \left( k + l\ CMF_{GLB} + m\ CMF_{GLB}^2 \right)$$

| Coefficient | a | b | c | d | k | l | m |
|---|---|---|---|---|---|---|---|
| Value | 3.8199 | -0.0081 | 1.4502 | -0.0076 | 0.4598 | -0.2771 | 0.0738 |

*Both the table and the equation were added to the text of the manuscript (Table 1 & pages 5– 6, lines 31, 1–3.*

6. Figure 5: It would help the reader, if you would use the same scale for each subfigure (e.g. [-50,50]) both to save space (just one color bar needed) and to enable comparison between subfigures.

*The actual values of the principal components between the figures are incomparable, they are only the artifacts of the different PCA analyses. Also, if the scale was to be unified to [-50,50], some of the subfigures would become illegible due to the relatively small range. Therefore, we have decided to keep the original figure. Another option, to avoid confusion, is not to include the numbers at all, but only to state "high" and "low" (see fig. res-2). This way the relative shapes of the geopotential heights fields created by the PCA are still fully visible.*

[Figure]

*Fig. res-2. An alternative option for Fig. 5, stating only the relative principal components' magnitude.*

7. The effect of SZA on radiation daily doses is trivial. I suggest to skip the analysis (table 4, second column and figure 3a) or to investigate the influence of surface albedo instead.

*Thank you for your comments. However, we confirmed that SZA is the most important factor affecting the daily doses; its effect is stronger than the effect of TOC or cloud cover. Therefore, we have decided to keep the analysis so that the effects of TOC and cloud cover can be compared to the very well-known effect of SZA. In addition, we have included the effect of surface albedo in fig. 3 and Table 4 (changed to Table 5 after the addition of Table 1). Please note that with the inclusion of surface albedo the partial correlation coefficients stated in the text (sect. 5.2) had to be changed according to the new regression model.*

8. Typos Page 7, line 8 "cover or" –> cover of

*Changed accordingly.*

==Anonymous Referee #2==

The paper discusses the reconstruction and analysis of 50-year time series of Erythemal UV radiation (UV-Ery) over the central European station Hradec Králové. The paper also addresses the connection of UV-Ery long-term changes (examining also the evolution of high UV-Ery doses) to large circulation patterns over Europe. The statistical methods applied are well suited for this type of analysis, and the paper contains new material. However, the paper needs minor improvements before accepted for publication.

The way the statistical methods are applied and the quantities used are not clearly described, and in some paragraphs may be even confusing. For example, in Page 1, line 18 (Abstract) it is written that "the number of days with very high EUV radiation increased by 22% per decade", a statement that cannot be easily deduced by either the description of methods (paragraph 4) or the discussion in the relevant paragraph (5.3).

Specific comments:

1. Page 3, lines 24-26: Is it possible to give an indication of the number of missing observations

   *Added to the text.*

2. Page 4, lines 7-8 (3.1.2 AOD): The meaning of this last sentence is not clear. Do you mean that the number of data available for the calculation of the AOD320 climatology was 61% of the total?

   *Yes, the time series was completed from 61 % (61 % of days had data). To avoid confusion, the formulation was changed accordingly in the text.*

3. Page 5, lines 25 – 30: Please give more details. Is there a reference you can give here? Last line (30-31): "To obtain the best fit function.." Where is this best-fit function used?

   *A reference for the Cloud Modification Factor calculation was added to the text (p. 5, l. 25–26).*
   *The use of multiple non-linear regression was also described in more detail, including the precise formula we used (p. 5–6, l. 31, 1–4).*

4. Page 6 line 1: Please change "The verification.." to "The validation.."

   *Changed accordingly.*

5. lines 3-4: "used to develop the model and the second..." Which model are you referring to here? LibRadtran? Please clarify.

*We mean the multiple non-linear regression model that was used to estimate the all-sky EUV irradiance. Clarification added to the text.*

6. line 30: "because 1995 was the year when the long-term ozone lowering stopped" It is better to write "because stratospheric (or total) ozone reached its minimum in 1995 over Europe"

*Changed accordingly.*

7. Page 7 line 8: typo "or" -> "of"

*Changed accordingly.*

8. line 8: "..therefore, low cloud cover was specified this way". Do you mean that the threshold for low cloud cover was set to 4 octas or less?

*Yes, clarified in the text.*

9. lines 10-15: Please give the period you used from the NCEP reanalysis

*The whole 50-years period 1964–2013, added to the text.*

10. line 15: Why do you use both indices? They are not independent from each other. See also comment below

*Answered together with comment no. 14.*

11. Page 8-9, Par. 5.2, and in connection to Figure 3. Please correct or discuss why this is done this way the figure. Are the correlation coefficients you found and discuss negative or positive? The text says negative, while the figure suggests otherwise.

*The correlation coefficients are of course negative in all three cases, fig. 3 was corrected.*

12. Page 9, paragraph 5.3: There is no clear description of the methods you used here, nor a clear description of what Figure 4 presents and how it was calculated. Here you discuss only days with high erythemal dose, not all available days. It seems that this is a result of partial correlation performed separately on the EUV90 or how was it done?

*The days with high erythemal doses ($EUV_{90+}$) were selected based on the $90^{th}$ percentile of each months, an explanation and the thresholds for each month are provided in sect. 4 (Table 3; p. 7, l. 7–9). The same section also describes the factors used to explain the days with $EUV_{90+}$ (p. 7, l. 9–14). The selected days were then explained based on the given thresholds of the factors and the number of days with $EUV_{90+}$ together with the explanatory factors was plotted into fig. 4. A cross-link to Methodology section was added to the text (p. 10, l. 10).*

13. How was the number referred to in the Abstract (22% per decade increase in the number of days) calculated?

*It was a linear trend analysis of the number of days with EUV90+ throughout the whole study period. This number, together with the error bars, was added to the text in sect. 5.3 (p. 10, l. 4–9).*

14. Page 10, lines 8 and below: Why do you use separately NAO and AO for your table 5 correlations?  Do you have an explanation as to their individual effects and why they should be examined separately? They are very closely connected, and there is no need to present both. It is better to discuss the effects as joined, as you do in the rest of the paragraph with the PCA analysis.

   *The NAO and AO indices are highly correlated but they are linked to slightly different processes in atmosphere. AO is more pole-centered and it is linked mainly to Arctic polar vortex, while NAO is related mainly to see-saw variability of the Icelandic Low and Azores High pressure regions. For this reason the influence of AO and NAO on Central European region may be different in different seasons. (Added to the text in Sect. 4, p. 7, l. 21–24, with relevant citations).*

**LIST OF RELEVANT CHANGES MADE IN THE MANUSCRIPT**

- Page 1, lines 16–17 (abstract): added a reference to surface albedo (new analysis suggested by Referee 1)
- Page 2, line 6 (Sect. 1): "radiation intensity" changed to "irradiance" (suggested by Referee 1)
- Page 3, line 24 (Sect. 3.1.1): added the number of missing values from the used TOC time series that had to be estimated based on reanalyzed datasets (suggested by Referee 2)
- Page 4, line 7 (Sect. 3.1.2): clarified formulation (suggested by Referee 2)
- Page 4, lines 13–17 (Sect. 3.1.3): clarification of surface global radiation albedo estimations using a neural networks ensemble (suggested by Referee 1)
- Page 5, line 17 (Sect. 3.2): clarification of the model solver used for the time series reconstruction (suggested by Referee 1)
- Page 5, lines 25–26 (Sect. 3.2): citation added (suggested by Referee 2)
- Pages 5–6, lines 31, 1–3 (sect. 3.2): clarification of the multiple non-linear regression model used for the Cloud Modification Factor (EUV) reconstruction; addition of the equation of the regression model (suggested by Referee 1 and Referee 2)
- Page 6, line 4 (Sect. 3.2): "verification" changed to "validation" (suggested by Referee 2)
- Page 6, lines 6–11 (Sect. 3.2): clarification of the data used to train and validate the multiple non-linear regression model (suggested by Referee 1 and Referee 2)
- Page 7, line 3 (Sect. 4): formulation changed (suggested by Referee 2)
- Page 7, line 4 (Sect. 4): added a reference to surface albedo (new analysis suggested by Referee 1)
- Page 7, line 12 (Sect. 4): typo corrected (suggested by Referee 1 and Referee 2)
- Page 7, lines 12–13 (Sect. 4): clarification of methodology (suggested by Referee 2)
- Page 7, line 17 (Sect. 4): study period specified (suggested by Referee 2)
- Page 7, lines 21–24 (Sect. 4): clarification why NAO and AO indices were used separately for the geopotential heights analysis (suggested by Referee 2)
- Page 9, lines 6–25 (Sect. 5.2): included the results of the analysis of the effect of surface UV albedo on EUV radiation doses (suggested by Referee 1)
- Page 10, lines 4–5 (Sect. 5.3): added the linear trend value and the relevant error bars (suggested by Referee 2)
- Page 10, line 9 (Sect. 5.3): added the linear trend value and the relevant error bars (suggested by Referee 2)
- Page 10, line 10 (Sect. 5.3): added a cross-link to Methodology section (suggested by Referee 2)
- Page 12, line 1 (Sect. 6): added a reference to surface albedo (new analysis suggested by Referee 1)
- Page 13, lines 24–25 (References): a new citation related to AO/NAO added (suggested by Referee 2)
- Page 17, lines 17–18 (References): a new citation related to AO/NAO added (suggested by Referee 2)
- Page 19 (Table 1): a new table with the multiple non-linear regression model coefficients added (suggested by Referee 1 and Referee 2)
- Page 23 (Table 5): included the results of the analysis of the effect of surface UV albedo on EUV radiation doses (suggested by Referee 1)
- Page 25 (Figure 1): "radiation" changed to "irradiance" (suggested by Referee 2)
- Page 27 (Figure 3): included the results of the analysis of the effect of surface UV albedo on EUV radiation doses (suggested by Referee 1)
- Page 27 (Figure 3): legend corrected (suggested by Referee 2)

[revised manuscript text omitted]

---

## Author Response (AR2)

Dear Dr Godin-Beekman,

Thank you for the suggestion, the manuscript was changed accordingly.

Kind regards,

Klara Cizkova

[revised manuscript text omitted]